# High-throughput targeted long-read single cell sequencing reveals the clonal and transcriptional landscape of lymphocytes

Mandeep Singh [1,2,4], Ghamdan Al-Eryani [1,2,4], Shaun Carswell[1], James M. Ferguson [1], James Blackburn [1,2], Kirston Barton[1,2], Daniel Roden[1,2], Fabio Luciani[2,3], Tri Giang Phan [1,2], Simon Junankar [1,2], Katherine Jackson [1,2], Christopher C. Goodnow [1,2,4], Martin A. Smith [1,2,4] & Alexander Swarbrick [1,2,4]

High-throughput single-cell RNA sequencing is a powerful technique but only generates short reads from one end of a cDNA template, limiting the reconstruction of highly diverse sequences such as antigen receptors. To overcome this limitation, we combined targeted capture and long-read sequencing of T-cell-receptor (TCR) and B-cell-receptor (BCR) mRNA transcripts with short-read transcriptome profiling of barcoded single-cell libraries generated by droplet-based partitioning. We show that Repertoire and Gene Expression by Sequencing (RAGE-Seq) can generate accurate full-length antigen receptor sequences at nucleotide resolution, infer B-cell clonal evolution and identify alternatively spliced BCR transcripts. We apply RAGE-Seq to 7138 cells sampled from the primary tumor and draining lymph node of a breast cancer patient to track transcriptome profiles of expanded lymphocyte clones across tissues. Our results demonstrate that RAGE-Seq is a powerful method for tracking the clonal evolution from large numbers of lymphocytes applicable to the study of immunity, auto-immunity and cancer.

[1] Garvan Institute of Medical Research, Sydney, NSW 2010, Australia. [2] St Vincent's Clinical School, Faculty of Medicine, UNSW, Sydney, NSW 2010, Australia. [3] Kirby Institute for Infection and Immunity, School of Medical Sciences, UNSW, Sydney, NSW 2052, Australia. [4] These authors contributed equally: Mandeep Singh, Ghamdan Al-Eryani, Christopher C. Goodnow, Martin A. Smith, Alexander Swarbrick. Correspondence and requests for materials should be addressed to C.C.G. (email: c.goodnow@garvan.org.au) or to M.A.S. (email: martinalexandersmith@gmail.com) or to A.S. (email: a.swarbrick@garvan.org.au)

Cell phenotypic diversity in humans and other vertebrates can arise from complex gene rearrangement and alternative RNA splicing events[1,2] that are not yet captured by current short-read RNA-sequencing technologies for measuring differential mRNA expression in single cells. A key example of this problem is the need for better ways to trace the response of single cells of the immune system during their response to cancer. Each newly-differentiated T or B lymphocyte in the immune system carries a different antigen receptor as the result of critical DNA rearrangements that alter the 450 nucleotides at the 5' end of their T- or B-cell antigen–receptor mRNA[3]. In the case of B lymphocytes, they use additional DNA rearrangements to 'isotype switch' between nine alternative constant region sequences comprising 1000–1500 nucleotides at the 3' end of the heavy chain mRNA[4], and use alternative mRNA splicing to change the 100–250 nucleotides at the 3' end of *IGH* mRNA in order to secrete the encoded receptors as antibody[5]. Similarly, complex gene rearrangements and alternative splicing events create pathological cell diversity amongst cancer cells[6]. Hence there is a critical need for methods that capture these sequence changes occurring throughout the length of mRNA molecules at single cell resolution, and integrate that information with gene-expression features.

The extraordinary diversity of antigen receptors on B and T lymphocytes governs the development, survival, and activation of these cells. T cells express on their cell surface a T-cell receptor (TCR) heterodimer composed of either α and β or γ and δ chains, each the product of a different germline *TRA*, *TRB*, *TRG*, or *TRD* gene locus, respectively. B cells express a B-cell receptor (BCR) hetero-tetramer composed of two identical membrane immunoglobulin heavy chains encoded by the *IGH* gene locus and two identical immunoglobulin kappa or lambda light chains encoded by the *IGK* or *IGL* genes, respectively. Each of these gene loci comprise in their germline configuration a cluster of separate variable (V), diversity (D), and joining (J) gene segments, one member of each cluster becoming joined through irreversible somatic DNA rearrangements during T or B lymphocyte development in a process known as V(D)J recombination[3]. Further diversity between cells is created by random addition or removal of nucleotides at the V(D)J junctions that encode complementarity determining region 3 (CDR3) in the antigen binding site of the receptor. The resulting diversity of the lymphocyte antigen–receptor repertoire is estimated at >10$^{12}$ different TCR or BCR proteins[7,8], governed by the rule of "one cell clone - one receptor sequence". Consequently, it is extremely unlikely that two cells descended from different lymphocytes will carry the same antigen–receptor sequence or 'clonotype'. As a result, when a B-cell or T-cell is stimulated by antigen to divide and undergo clonal expansion, the BCR or TCR sequence serves as a unique 'clonal barcode' and provides information on antigen specificity and cell ancestry.

Sequencing the BCR or TCR of individual lymphocytes in parallel with their transcriptome provides high-resolution insights into the adaptive immune response in a range of disease settings such as infectious disease, autoimmune disorders, and cancer[9,10]. A common approach to link paired antigen–receptor sequences with gene-expression profiles of single lymphocytes is through the use of the full-length single-cell RNA-Sequencing (scRNA-Seq) method Smart-Seq2[11], where computational methods can reconstruct paired TCRα and TCRβ sequences or paired heavy and light chain sequences from Illumina short-reads[12–16]. However, Smart-Seq2 generally relies on plate- or well-based microfluidics and is therefore limited in the number of cells that can be processed, typically 10–100 s. Additionally, a large number of sequencing reads are generally required to computationally reconstruct paired antigen receptors[17]. As such, the cost per cell is

relatively high, estimated at \$50–\$100 USD[18]. Moreover, assembly of short reads makes it difficult or impossible to decipher critical alternative splicing of mRNA segments separated by more than 500 nucleotides, as occurs in *IGH* genes.

Recent technological advancements in high-throughput scRNA-Seq methods allow thousands of cells to be captured and sequenced in a relatively short time frame and at a fraction of the cost[18]. Such methods rely on capture of polyadenylated mRNA transcripts followed by cDNA synthesis, pooling, amplification, library construction, and Illumina 3' cDNA sequencing[19–26]. The combination of fragmentation and short-read sequencing fails to sufficiently sequence the V(D)J regions of rearranged TCR and BCR transcripts, which are located in the first 500 nucleotides at the 5' end of the transcript. Consequently, 3'-tag scRNA-Seq platforms have limited application for determining clonotypic information from large numbers of lymphocytes. Variations on this approach employing 5' cell barcodes enable the V(D)J sequences and global gene expression to be measured[27], but don't solve the need to integrate this information with the diversity of switching and alternative mRNA splicing involving the 3' end of *IGH* mRNA. Recent advances in long-read sequencing technologies present a potential solution to the shortcomings of short-read sequencing. Full-length cDNA reads can encompass the entire sequence of BCR and TCR transcripts, but typically suffer from higher error rates and lower sequencing depth than short-read technologies[28].

Here, we describe a rapid high-throughput method to sequence full-length transcripts using targeted capture and Oxford Nanopore sequencing and link this with short-read transcriptome profiling at single cell resolution. This novel method, termed Repertoire and Gene Expression by Sequencing (RAGE-Seq), can be applied to high-throughput droplet-based scRNA-Seq workflows to accurately pair gene-expression profiles with targeted full-length mRNA sequences from a large number of cells. We demonstrate the power of this method by combining transcriptome profiling with full-length antigen–receptor sequence characterization from thousands of human tumor-associated lymphocytes. Using de novo assembly of nanopore reads, complete antigen–receptor sequences can be recovered at high accuracy and sensitivity, including the identification of somatic mutations from immunoglobulin full-length heavy and light chains allowing the inference of B-cell clonal evolution.

## Results

**RAGE-seq workflow**. To integrate short-read and long-read mRNA sequence analysis of thousands of single cells, we designed a strategy to split full-length single-cell 3'-tag cDNA libraries prior to fragmentation for short-read sequencing, and selectively enrich BCR and TCR cDNA transcripts using targeted hybridization capture. Targeted capture was chosen over more commonly used PCR methods for repertoire analysis[29,30] so that full-length transcripts were retained. Enriched antigen-receptor molecules are then subjected to long-read Oxford Nanopore sequencing to obtain both the 3' cell-barcode and the 5' V(D)J sequence. In parallel, short-read Illumina sequencing to profile gene expression is conducted on the remaining cDNA (Fig. 1 and Supplementary Fig. 1a). By matching the cell barcodes obtained from long-read sequencing with the cell barcodes obtained from short-read sequencing, transcriptome profiles for each individual cell can be linked with full-length antigen–receptor sequences.

We designed a capture bait library with probes specifically targeting all annotated and functional human V, J, and constant region exons within the genomic loci that encode all TCR and BCR chains. Whole-genome assemblies generated from long-read sequencing often use de novo assembly followed by 'polishing' to

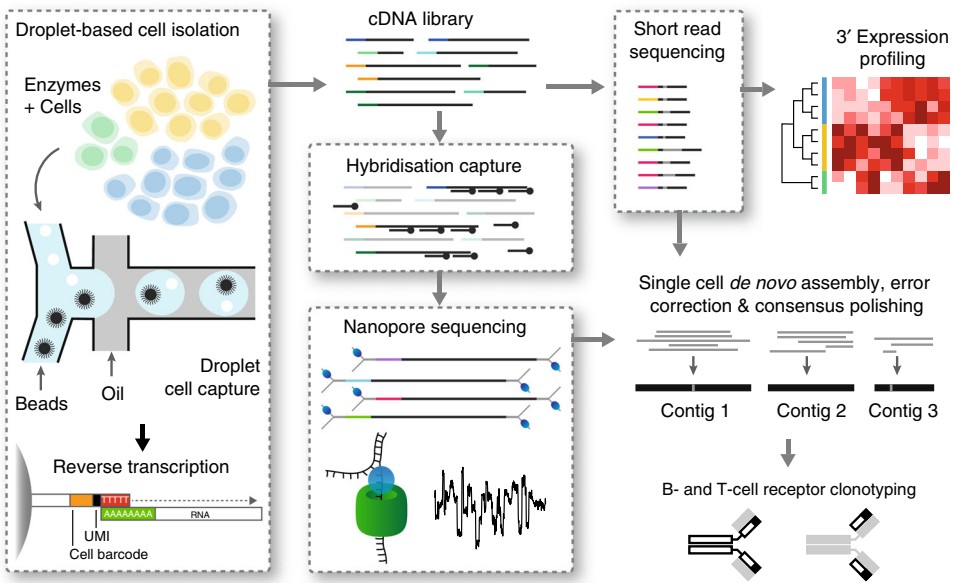

**Fig. 1** Overview of RAGE-Seq. Droplet-based scRNA-Seq is used to generate an initial barcoded cDNA library, which is split and simultaneously subjected to (i) short-read sequencing for 3′ expression profiling and (ii) targeted capture using custom probes followed by long-read sequencing. The short-read sequencing is used to generate highly accurate cell-barcode sequences which permit demultiplexing of the long-read data. Demultiplexed long-reads are subjected to de novo assembly and error correction to generate full-length BCR and TCR mRNA sequences, with single nucleotide accuracy. Transcriptome profiles generated from short-read sequencing can then be linked to the antigen–receptor sequence for each individual cell

achieve high accuracy over 99%. We predicted that such approaches could also be applied to nanopore reads generated from cDNA-targeted capture and developed a computational pipeline that combines de novo assembly with clonotype assignment on demultiplexed nanopore data to generate full-length TCR or BCR sequences for each cell (Supplementary Fig. 1b). We chose de novo assembly over an alignment-based approach since the low accuracy of individual nanopore reads would be problematic for alignment to antigen–receptor loci, which contain hundreds of small interspersed exons.

**Cross-platform sequencing validation.** To assess the validity of our method we performed RAGE-Seq on a mixture of the human T-cell line Jurkat and the human B-cell line Ramos, for which antigen–receptor sequences are published[31,32] (Supplementary Fig. 2a). A proportion of ~15% human monocyte cells were added to serve as a negative control. The dataset consisted of 1463 Jurkat cells, 2000 Ramos cells, and 280 monocytes (Fig. 2a and Supplementary Fig. 2b).

Following nanopore sequencing, a total of 20,346,396 reads were obtained, 42.9% of which uniquely aligned to TCR and BCR constant regions (on-target reads), representing a ~13-fold enrichment when compared to non-targeted capture Illumina data (Supplementary Fig. 2c). Demultiplexing of nanopore reads with cell barcodes identified by short-read sequencing yielded 18.7% of total nanopore reads containing complete recovery of barcode library sequences (Fig. 2b and Supplementary Table 1). It is noteworthy to mention that only 77.2% of the Illumina reads were used to demultiplex the long reads, as the remaining Illumina barcodes were associated with low read counts, indicative of unproductive cell capture (Supplementary Table 2). Barcode recovery for nanopore reads that were on-target was 99.3 and 100% for Jurkat and Ramos cells, respectively (Supplementary Fig. 2d). A strong correlation of the abundance of T-cell receptor alpha constant gene (TRAC) reads per cell between Oxford Nanopore and Illumina sequencing was also observed (Pearson correlation = 0.79, Fig. 2c). These results demonstrate

that the amplified full-length cDNA library can be sufficiently sampled between the two sequencing platforms.

**Identification of antigen–receptor sequences.** To generate accurate antigen–receptor sequences for each cell we carried out de novo assembly using Canu[33] followed by two consecutive rounds of assembly polishing with Racon[34] and Nanopolish[35] (see Methods). Each assembly comprised on average 4.26 contigs per Jurkat cell, 5.24 per Ramos cell and 0.12 per monocyte (Supplementary Fig. 3a). On average, 30% of contigs for Jurkat cells and 32.9% of Ramos cells were assigned a productive antigen–receptor sequence. The nucleotide length of the contigs assigned antigen–receptor sequences were consistent with the predicted full-length Jurkat TRA and TRB and Ramos IGH and IGK reference mRNA transcripts (Fig. 2d). Importantly, this shows that our de novo assembly approach can retain full-length mRNA transcripts.

For Jurkat cells, we recovered 18.9% of cells with full-length mRNA contigs encoding paired TCRα and TCRβ chains, 13.3% with a TCRα chain only and 39.6% with a TCRβ chain only. For Ramos cells, we recovered 31% of cells with contigs encoding the full coding regions for paired immunoglobulin heavy and light chains, 33% with a heavy chain only and 19.1% with a light chain only (Fig. 3a and Supplementary Fig. 3b,c). There was little assignment of non-reference V and J genes (Supplementary Fig. 3d,e).

Next, we evaluated the accuracy of calling a correct clonotype at nucleotide resolution by investigating the CDR3 region of Jurkat cells against their known reference CDR3 sequences. Of Jurkat cells with an assembled TRA or TRB contig, the percentage with the correct reference CDR3 sequence was very high. 98.9% expressed the reference CDR3α sequence and 99.6% expressed the reference CDR3β sequence while the number of cells carrying non-productive sequences was small (Fig. 3b). Assembly polishing was found to modestly increase the recovery of cells with productive chains (3.15% for TCRα and 6.14% for TCRβ) and had a small effect on the overall accuracy (Supplementary Fig. 3f).

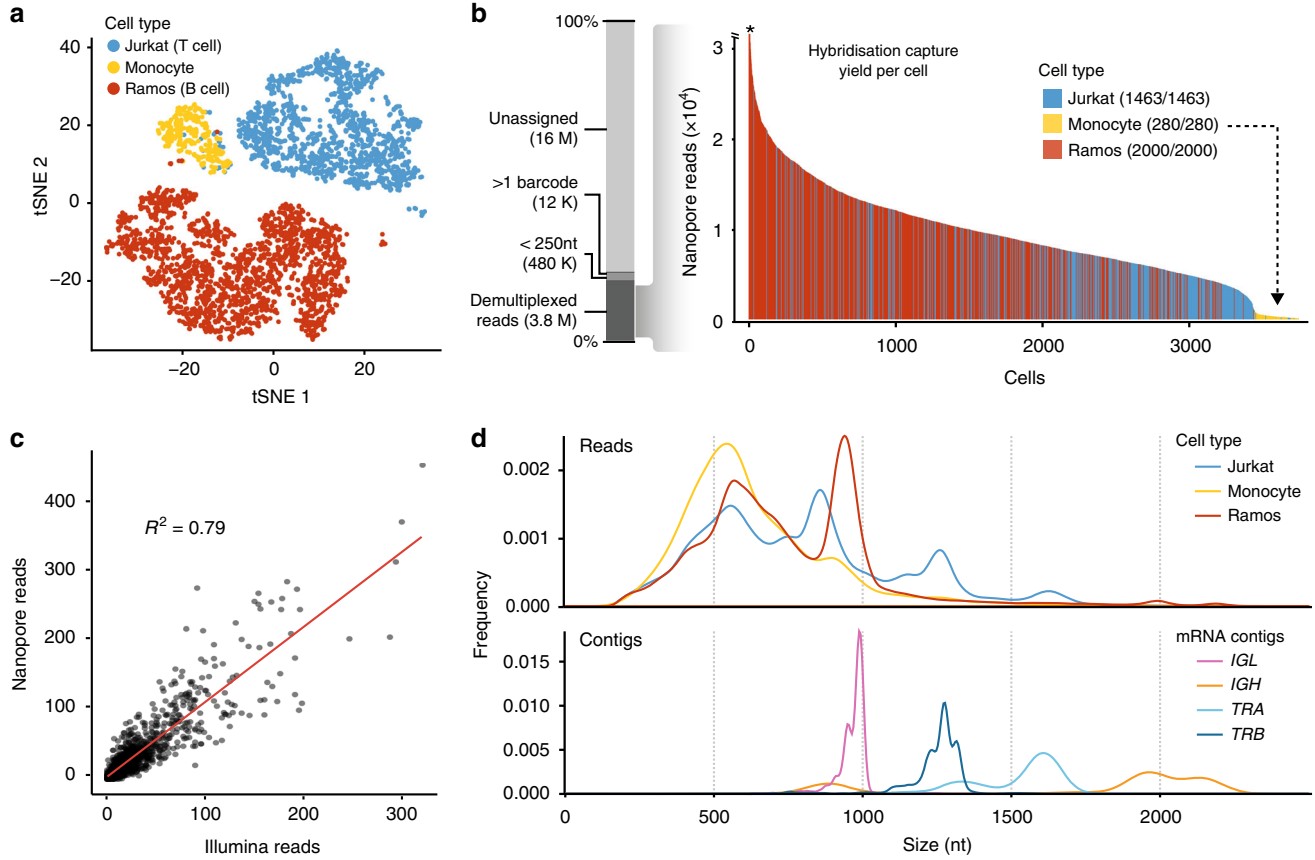

**Fig. 2** Short-read and targeted long-read single cell sequencing of immortalised B and T cell lines. **a** T-distributed stochastic neighbour embedding (t-SNE) analysis of single cells generated from short-read sequencing data (number of cells: Jurkat = 1463; Ramos = 2000; monocytes = 280). **b** Demultiplexing statistics for nanopore sequencing reads following targeted capture. Each bar corresponds to the number of nanopore reads per cell barcode identified with short-read sequencing using exact sequence matching. Asterix indicates one cell with over 6000 reads. The number of recovered cell barcodes is shown next to each cell type. " >1 barcode" refers to more than one cell barcode found in a single read and " < 250 nt" refers to any read shorter than 250 nt. **c** Correlation between Illumina read counts and Oxford Nanopore read counts for T-cell receptor alpha constant gene (*TRAC*). Each point represents an individual Jurkat cell (*n* = 472). Pearson correlation = 0.79. **d** Nanopore read length distribution of demultiplexed reads assigned to each cell type (top panel) compared to the length distribution of assembled contigs that have been assigned a productive receptor chain (bottom panel). Predicted lengths (nt) of mRNA transcripts: Jurkat *TRA*, 1552 nt; Jurkat *TRB*, 1,259 nt; Ramos *IGH* (secreted exons), 1485 nt, Ramos *IGH* (membrane exons), 1683 nt; Ramos *IGL*, 932 nt. Predicted lengths obtained from the IMGT database[60]

We also found that read depth impacted the total number of contigs recovered (Fig. 3c), but had little effect on the CDR3 accuracy for both Jurkat TCRα and TCRβ chains and Ramos immunoglobulin heavy and light chains (Fig. 3d).

We also compared RAGE-Seq against the reconstruction of TCR sequences from 28 Jurkat cells produced using Smart-Seq2 and VDJPuzzle[13]. VDJPuzzle was able to recover a greater percentage of cells with a receptor chain: 22/28 Jurkat cells were assigned a TCRα chain and 25/28 Jurkat cells were assigned a TCRβ chain. The throughput of RAGE-Seq, however, was much greater for an experiment of a comparable timescale and proved to be ~23 times more cost effective on a per cell basis (Supplementary Tables 3 and 4). Taken together, these results indicate that RAGE-Seq is both accurate and sensitive in determining clonotype sequences and has significant advantages over Smart-Seq2 in terms of cost and throughput.

**B-cell clonal network analysis**. The Ramos cell line is known to mutate its receptors by undergoing somatic hypermutation in culture[36] at a reported rate of 0.84 mutations in the *IGHV* gene per generation[36]. To identify point mutations resulting from somatic hypermutation in individual B cells, accurate sequence characterization across the entire V(D)J region of mRNA

encoding heavy and light chains is required. RAGE-Seq was able to recover 98.5% of Ramos cells with complete *IGHV* sequences and 98.8% of Ramos cells with complete *IGLV* sequences (Supplementary Fig. 4a). We determined amino acid replacement mutations spanning full-length V regions of the heavy and light chain from 615 Ramos cells assigned paired chains. Conserved amino acid mutations were observed in six different heavy and light chain positions and a dominant subclone within the Ramos cell line was found at a frequency of 147/615 cells along with 37 subclones represented by more than one cell and 319 subclones represented by a single cell (Fig. 4a). We generated a clone network based on nearest neighbor distance using the inferred germline sequence as the unmutated ancestor, demonstrating the evolution of individual Ramos cells undergoing active somatic hypermutation (Fig. 4b). Thus, RAGE-Seq can pair transcriptomic phenotype to the evolution of immunoglobulin sequences of individual B cells within clonal populations.

To assess the accuracy of calling point mutations we investigated Jurkat *TRAV* and *TRBV* genes, which should be completely conserved in this clonal cell line. We found only a low number of Jurkat cells with one or more inferred nucleotide mismatches to germline in these regions (Supplementary Fig. 4b). We measured the mutation rate of Jurkat *TRAV* and *TRBV* genes

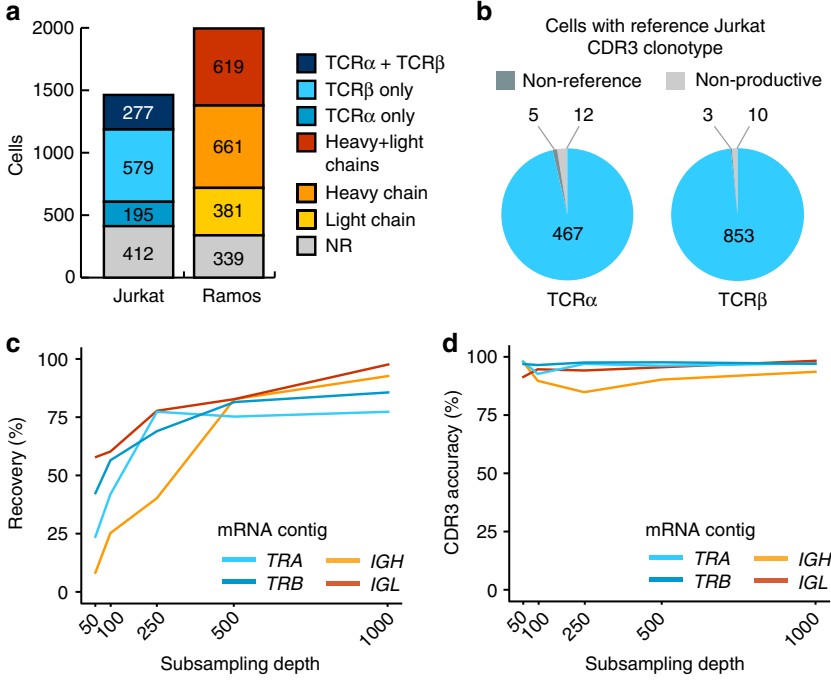

**Fig. 3** Validation of antigen–receptor assembly. **a** Number of cells assigned productive TCRα and TCRβ chains for Jurkat cells (*n* = 1463) or productive heavy and light chains for Ramos cells (*n* = 2000). Only receptor chains expressing the reference V and J gene combinations of Jurkat (*TRA: TRAV8-4, TRAJ3*; *TRB: TRBV12-3,* and *TRBJ1-2*) or Ramos (*IGH: VH4-34, IGHJ6; IGL: IGLV2-14,* and *IGLJ2*) were assigned. NR no receptor. **b** CDR3 accuracy measured by the number of Jurkat cells with assigned *TRA* or *TRB* sequences that directly match the reference Jurkat CDR3 nucleotide sequences (Supplementary Fig. 2a). 'Non-productive' refers to a cell with a CDR3 sequence that is out-of-frame or contains stop-codons. 'Non-reference' refers to a cell with a productive CDR3 sequence that does not match the reference. Only cells with Jurkat reference V and J gene combinations were analysed. **c** Recovery of TCR and BCR chains as a function of sequencing depth. Subsampling was performed on exactly 200 Jurkat cells and 200 Ramos cells with >1000 nanopore reads and assigned paired receptor chains. For Ramos, cells with the most common *IGH* and *IGL* CDR3 sequence were pre-selected as the reference sequence (Supplementary Fig. 2a). Subsampling was performed at the indicated read depths on the X-axis. **d** Accuracy of the assembled CDR3 sequence as a function of sequencing depth, as described in **c**. The percentage of cells with a CDR3 sequence that matched the reference CDR3 sequence was measured at each subsampling depth

across all cells at 0.095% and 0.032%, respectively. In contrast, the mutation rate of Ramos *IGHV* and *IGLV* genes was 3.41% and 2.13%, respectively. These results suggest that RAGE-Seq can accurately identify somatic hypermutation of BCRs with low background.

**Analysis of lymphocytes from a human lymph node**. To apply our method to primary B and T lymphocytes, we performed RAGE-Seq on a tumor and paired lymph node resected from a triple negative breast cancer patient. In the lymph node, we identified 4165 T cells that could be subdivided into seven T-cell populations (Fig. 5a and Supplementary Fig. 5a). We recovered 705 (16.9%) T cells with paired TCRα and TCRβ chains, 1199 (28.7%) cells with a TCRα chain only and 762 (18.3%) cells with a TCRβ chain only. The recovery rate of TCR chains was comparable across the different T-cell subsets (Fig. 5b). We could also detect two different *TRA* or *TRB* sequences in 138 (9.5%) and 35 (1.8%) T cells, respectively, a frequency similar to previous reports[9,11]. Among the 1619 B cells in the lymph node, we recovered 689 (42.6%) cells with paired immunoglobulin heavy and light chains, 188 (11.6%) cells with only a heavy chain and 557 (35.6%) cells with only a light chain (Fig. 5b). Similar to the cell line experiment, all the cell barcodes were recovered across both sequencing platforms and full-length receptor chain sequences were assembled (Supplementary Fig. 5b–d).

Naïve B cells predominantly co-express *IGH* mRNAs with identical V(D)J sequences at their 5′ end but different constant region sequences at the 3′ end, produced by alternative mRNA

splicing of the V(D)J exon to either *IGHM* or *IGHD* exons[37]. By contrast most memory B cells have undergone isotyping class switching to *IGHA* or *IGHG* constant regions. We used the gene-expression data to classify naïve B cells as *IGHD + IGHG- IGHA-* and memory B cells as *IGHD- IGHG + IGHA +* (Supplementary Fig. 5A). In the breast cancer lymph node, most cells classified as naïve by gene expression had *IGHM* or both *IGHM* and *IGHD* antigen–receptor mRNA transcripts by nanopore sequencing while more than two-thirds of memory B cells expressed *IGHA* or *IGHG* (Fig. 5c). The fact that *IGHD* was not detected in many *IGHM*-bearing naïve cells is consistent with *IGHM* mRNA being 10-fold more abundant than *IGHD* mRNA[38]. Upon activation, B cells acquire point mutations in their V(D)J exon through somatic hypermutation[4]. Memory B cells in the lymph node had a mean mutation rate of 4.96% in their V segment heavy chain mRNA across all isotypes, consistent with published reports[14,39]. By contrast, naïve B cells had a much lower rate of 0.35% with mutations limited to a small number of cells that likely correspond to unswitched memory B cells (Fig. 5c and Supplementary Fig. 5e). These data demonstrate the ability of RAGE-Seq to detect class switching and somatic hypermutation in human tissue.

For each *IGH* transcript isotype, alternative splicing at the 3′ end generates either membrane-bound or secreted forms of immunoglobulin, with the latter gradually predominating as activated or memory B cells differentiate into antibody-secreting plasmablast cells[40]. The presence of shared V(D)J sequences in both membrane and secretory *IGH* isoforms was detected in

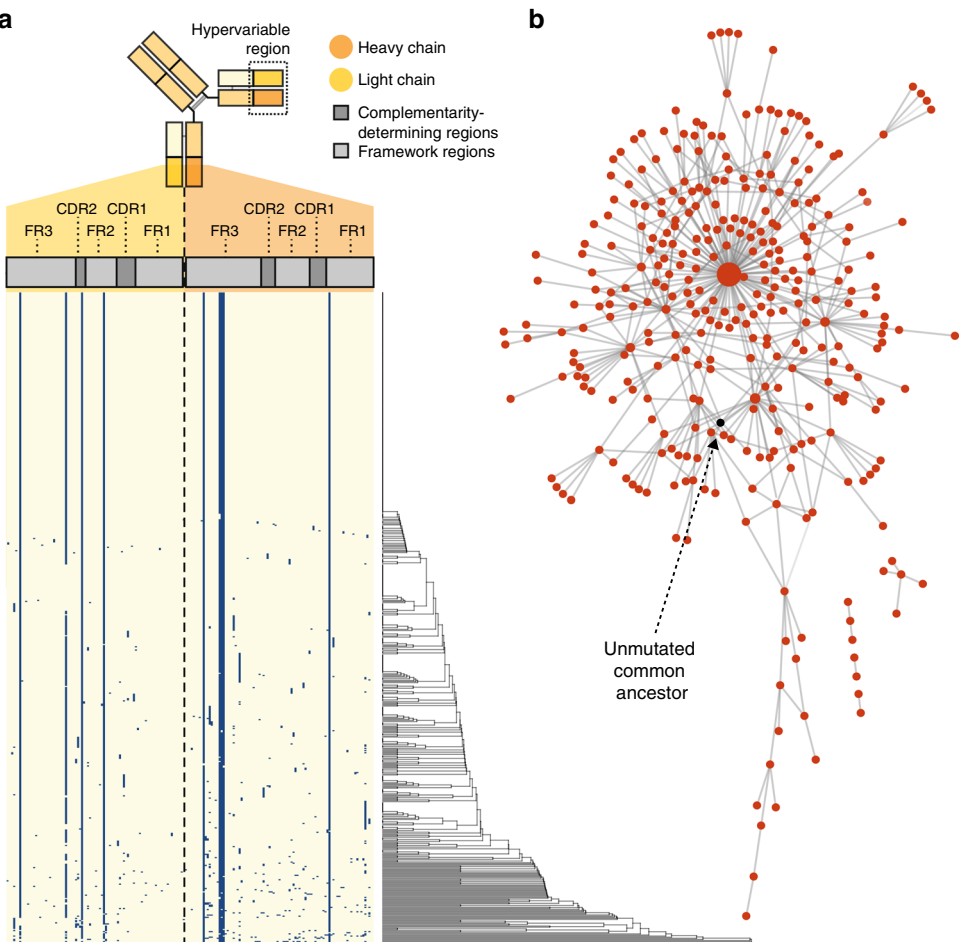

**Fig. 4** Tracking somatic hypermutation in an immortalized B-cell line. **a** Amino acid composition of the heavy and light chain V regions of individual Ramos cells assigned paired BCRs ($n = 615$). Each row represents an individual cell and each column a single amino acid position. Positions that are blue represent an amino acid that differs to the germline sequence, indicative of somatic hypermutation. On the right, a hierarchical clustering dendrogram of the concatenated heavy and light chain V region amino acid sequences is shown. **b** Network diagram of individual Ramos cells undergoing somatic hypermutation from **a**, where each node corresponds to a unique full-length heavy and light chain V(D)J sequence and the edges correspond to the number of amino acid differences between them. The largest node in the centre is the predominant sequence in the Ramos cell line represented by 147 cells. The unmutated common ancestor sequence in black was inferred from germline V(D)J sequences and is not represented by any cells in the dataset. Network diagram generated with Cytoscape[66]

many single naïve and memory B cells (Fig. 5c), consistent with previous evidence from pooled naïve cells[38]. As expected, only the secretory spliced form was detected in plasmablasts in the tumor from the same patient (see below and Supplementary Fig. 6a–c), and most were assigned *IGHA1* isotypes (Fig. 5c). This is consistent with plasmablasts and plasma cells having differentiated into high-rate antibody-secreting cells, and with the dominant switching to IgA in plasma cells of normal and neoplastic breast tissue[41].

Our targeted capture panel included probes against *TRG* and *TRD* genes allowing for the detection of γδ T cells, a poorly-explored class of unconventional T cells of substantial interest to studies of infection and tumor immunology[42]. A total of 11 T cells in the lymph node were assigned paired TCRγ and TCRδ chains, the majority of which clustered in the CD8 effector cluster. We also recovered 92 T cells with only the TCRγ chain and 14 T cells with only the TCRδ chain, again the majority clustering in the CD8 T-cell effector population (Fig. 5d). T cells assigned TCRγ chains alone were found to frequently co-express TCRα and TCRβ chains, consistent with the timing of TCRγ rearrangement[43]. In contrast, T cells assigned paired TCRγ and TCRδ chains were not co-assigned TCRα or TCRβ chains

suggesting that they are committed γδ T cells (Supplementary Table 5). We explored the identification of other unconventional T cells that can recognise non-peptide antigens based on their invariant TCR usage such as Mucosal Associated Invariant T (MAIT) cells[44] and Germline-Encoded Mycolyl lipid-reactive (GEM) T cells[45]. Ten T cells were found to carry MAIT-associated TCRs which clustered closely together in the CD8 effector population, while two T cells with GEM-associated TCR chains were found in the CD4 effector memory cluster (Fig. 5d). Interestingly the T cells carrying MAIT-associated TCRs all comprised of a single expanded clone (Supplementary Fig. 5f).

Evidence for substantial clonal expansion in the lymph node was uncommon, with shared V(D)J sequences only detected in pairs of cells. For B cells assigned paired BCR chains there were 13 cell pairs with the same BCR sequence, the majority of which segregated in the naïve B-cell cluster, while for T cells assigned paired TCR chains there were also 13 cell pairs with shared TCR sequence that also clustered by cell type (Fig. 5e). B-cell and T-cell clones with the same receptor sequence presented more similar gene-expression profiles than non-clonally expanded B cells ($P = 2.10E-07$, paired Wilcoxon test) and T cells ($P = 2.55E-11$) when comparing their Jaccard similarity coefficient for the 250 most

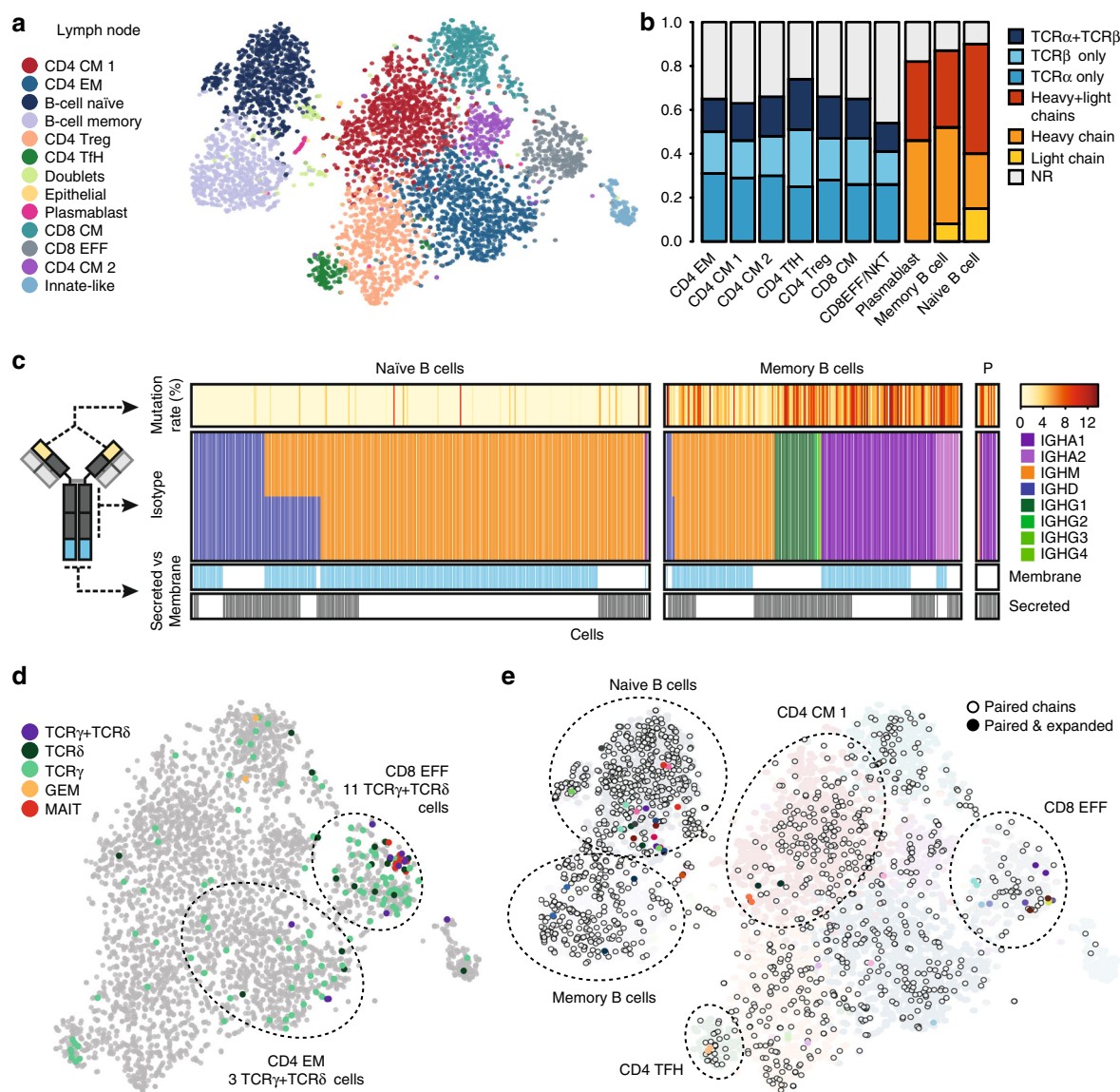

**Fig. 5** RAGE-Seq on a human lymph node. **a** t-SNE analysis of 6027 lymph node cells generated from short-read sequencing data. Number of cells: B-cell naïve, 853; B-cell memory, 738; CD4 effector memory (EM), 1069; CD4 central memory (CM) 1, 1069; CD4 CM 2, 226; CD4 T follicular helper cell (TfH), 142; CD4 T regulatory cell (Treg), 740; CD8 CM, 487; CD8 effector (EFF), 405; Plasmablast, 28; Innate-like, 144; Doublets, 86; Epithelial, 13. **b** Assignment of productive TCR and BCR chains to each population identified in **a**. NR, no receptor. **c** Characterization of full-length *IGH* mRNA sequences assigned to individual naïve (*n* = 401) or memory B cells (*n* = 283) from the lymph node or plasmablasts (P, *n* = 15) from a matched tumor (Supplementary Fig. 6). Mutation rate (%) measures the percentage of nucleotides in the V region mutated from germline. **d** Assignment of TCRγ chains, TCRδ chains and invariant TCR chains associated with MAIT and GEM T cells to the T-cell populations of the lymph node in **a**. 92 T cells were assigned TCRγ chains alone, 14 T cells were assigned TCRδ chains alone and 11 T cells were assigned paired TCRγ and TCRδ chains. 10 T cells were assigned MAIT-associated TCR chains and two T cells assigned GEM-associated TCR chains (see Methods). **e** Visualisation of the lymph node t-SNE plot in **a** for cells assigned paired BCR (*n* = 689) or paired TCR (*n* = 705) chains and amongst these cells those clones that are expanded. Clones were considered expanded if a paired TCR or BCR sequence was found in more than one cell. Different colors denote each expanded clone. 13 T-cell expanded clones and 13 B-cell expanded clones were identified. Each clone was represented by two cells

abundant genes chosen from within each respective cell type cluster (Supplementary Fig. 5g).

**Analysis of lymphocytes across tissues**. An important application of RAGE-Seq is the ability to track clonally related T or B cells across tissues, to gain systems-level insights into the evolution of immune responses. One such application is the analysis of lymphocytes in a tumor and its draining lymph node, the presumptive site of antigen presentation and source of tumor-infiltrating lymphocytes (TILs). In parallel with the lymph node analysis above, we performed RAGE-Seq on the patient's resected

primary breast tumor. From a total of 2493 captured cells, 909 T cells and 215 B cells were identified (Supplementary Fig. 6a–c). A proportion of receptor chains were found to be shared by lymphocytes within each tissue: 32/1143 light chains amongst 157 B-cells and 11/1771 TCRα chains, 7/1475 TCRβ chains and 5/155 TCRαβ chains amongst 134 T cells (Fig. 6a). Some chains showed significant tissue-specific enrichment, with one *IGL* sequence (V: *IGLV4-69*, J: *IGLJ3*, CDR3: QTWGTGFWV) expressed by 27 tumor-resident B cells and plasmablasts (16.9% of all light chains in tumor), but undetected in the lymph node (Supplementary Data 1).

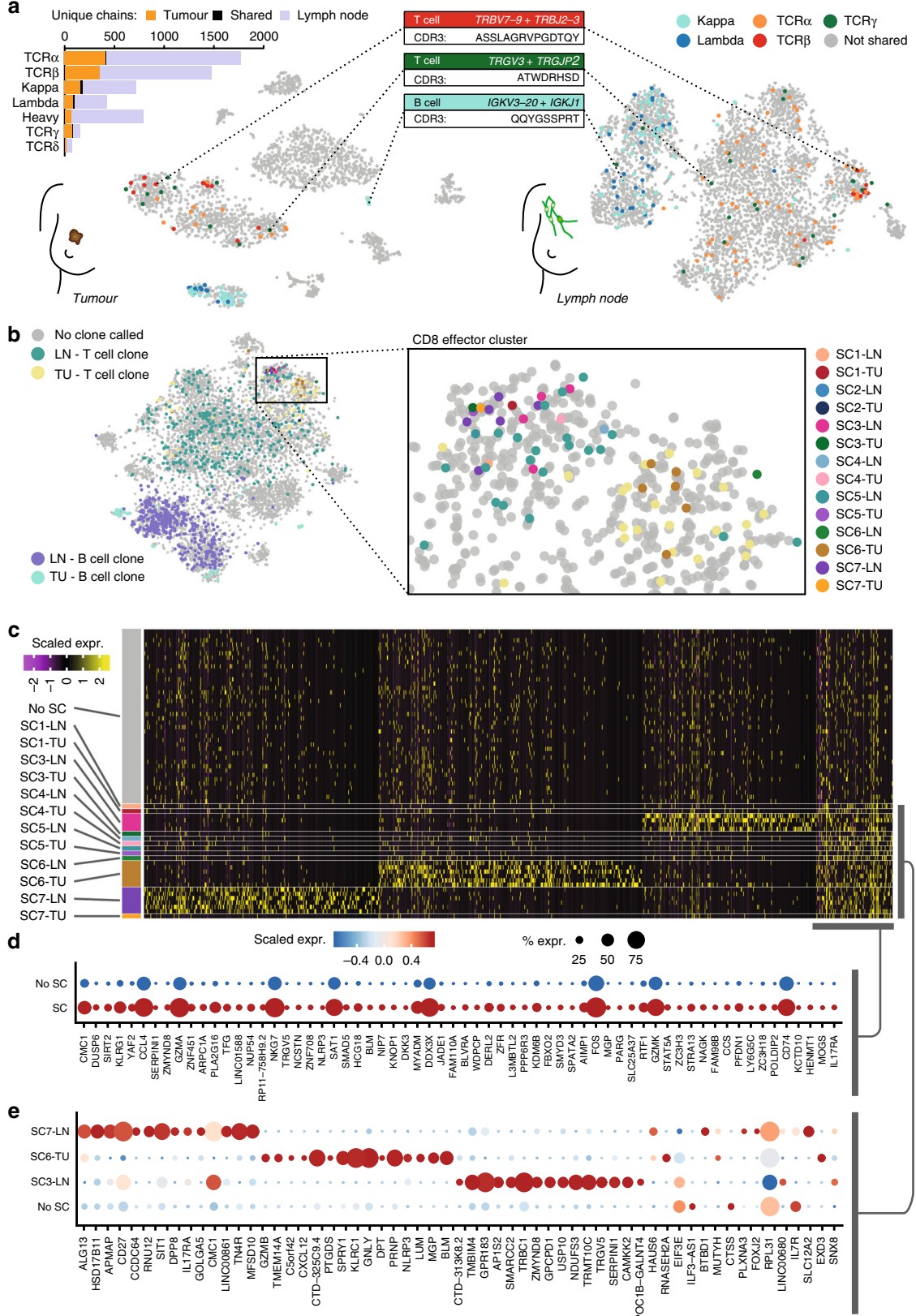

To investigate whether clonally related lymphocytes across the tumor and lymph node have common gene-expression features, we analyzed T-cell clones expressing identical TCRβ chains found across both tissues (Supplementary Data 1). Integration of tumor and lymph node gene-expression datasets revealed that cells belonging to six out of seven TCR clonotypes common between tissues clustered within the CD8 T-cell effector cluster, suggesting that TIL clones can maintain the same 'cell state' across lymph node and tumor (Fig. 6b). Differential gene-expression analysis between cells of the same clone within each tissue site did not

**Fig. 6** Tracking lymphocytes across a matched lymph node and tumor. **a** t-SNE analysis of patient matched tumor (n = 2493, see Supplementary Fig. 6) and lymph node (see Fig. 5). Cells expressing a shared receptor chain sequence found in both the tumor and lymph node datasets are highlighted and grouped by receptor chain type. The most frequent TCRβ (n = 10), TCRγ (n = 13) and immunoglobulin light chain (n = 20) sequence is highlighted. **b** Integrated t-SNE analysis of tumor and lymph node datasets (see Methods, n = 8,520). Cells assigned paired TCR chains or paired immunoglobulin heavy chains are highlighted. Seven TCR chain sequences found in both tumor and lymph node datasets (single chains; SC) are shown (across n = 30 cells) and six (every clonotype except SC2) are found in the highlighted box containing the CD8 effector cluster of both lymph node and tumor datasets. **c** Heat-map of differentially expressed genes (n = 1,328; P < 0.01, Wilcoxon signed-rank test) within the CD8 effector cluster highlighted in **b**. 50 'non-shared cells' were randomly chosen for visualisation purposes. **d, e** Dotplot illustrating the top 65 genes differentially expressed between **d** all shared clonotypes (SC) and non-shared clonotypes (No SC) or **e** the top three most frequent shared clonotypes (SC7, SC6, and SC3) and non-shared clonotypes. LN lymph node, TU tumor, SC shared clonotype

reveal any significant differences; however, analysis of cells within the same clone across tissues revealed differences for the three most frequent clones (SC3, SC6, and SC7) (Fig. 6c). Each TCR clonotype found across both tissue sites shared certain gene-expression features, including elevated expression of genes characteristic of active tissue resident cytotoxic lymphocytes, such as *CCL4*, *NKG7*, *GZMA*, and *GZMK*[46], suggesting these clones were activated in response to antigen (Fig. 6d). However, differential gene-expression analysis between clones revealed that each clone also has unique gene-expression features (Fig. 6e), suggesting a distinct differentiation state for each clonotype.

The presence of clonally expanded T cells between tissues suggested that these cells were proliferating in response to antigen stimulation. To examine this further, we used the scRNA-Seq data to perform cell cycle analysis of all cells within each CD8 T-cell effector cluster of tumor and lymph node to infer whether TIL persistence of the clone is through proliferation occurring at the site of each sample, or through trafficking between tissues. A large proportion of T cells were in S, G2, or M phase, suggesting ongoing proliferation. Interestingly, proliferation of expanded clones in the tumor was found to be comparable to non-expanded clones (Supplementary Fig. 6d,e).

## Discussion

Single cell methods are needed to capture cell diversity arising at both ends of individual mRNA molecules, such as pairing the 5′ clonotype-specific V(D)J sequence of BCR or TCR transcripts with different 3′ sequences for secreted or membrane forms of different immunoglobulin isotypes, and with global expression profiling of 3′ ends. Here, we report a generalisable experimental workflow and computation pipeline to integrate single cell gene expression with targeted characterization of full-length mRNA transcripts.

We have found RAGE-Seq to be robust in its ability to sample across both Illumina and Nanopore sequencing platforms and highly sensitive and accurate in providing full-length BCR and TCR sequences across immortalized and primary human B and T cells. Given its greater throughput and substantially lower cost, RAGE-Seq has significant advantages over other methods for immune profiling. As a result, RAGE-Seq can circumvent the need to isolate specific lymphocyte populations by flow cytometry, permitting retrospective characterization of low abundance lymphocytes within tissues. We were able to identify clones with unique gene-expression features that had expanded and were shared across tissues, despite unbiased sampling from a breast cancer, which generally have low TIL frequency[47]. The capacity to screen large numbers of lymphocytes could have significant translational applications. Response to checkpoint inhibitors for immunotherapy has been linked to TCR clonality[48], TIL frequency[49], and gene-expression signatures[50], yet these biomarkers have not been integrated at the single cell level. The application of RAGE-Seq to biopsies collected prior to and following treatment may accelerate the discovery of biomarkers or cell states that

predict response to therapy. Additionally, the recovery of paired antigen receptors can be used with complimentary approaches to identify TCR ligands[51].

In this study we demonstrate the compatibility of RAGE-Seq with the 10x Genomics Chromium 3′ system. RAGE-Seq is also adaptable to any high-throughput scRNA-Seq technologies that employ 3′ or 5′ cell-barcode tagging[20–25]. Two recently developed high-throughput droplet-based scRNA-seq methods also report antigen–receptor sequences: the commercially available 10x Genomics V(D)J + 5′ Gene Expression kit and DART-Seq[52]. Compared to these two methods, RAGE-Seq has several advantages (Supplementary Table 6), although the estimated cost per cell is marginally higher than the 10 × 5′ V(D)J kit. Most importantly, RAGE-Seq provides full-length receptor sequence and reports BCR somatic hypermutation. This is greatly beneficial for the analysis of immunoglobulin somatic hypermutation and for the synthesis of recombinant antibodies, which can be used to explore the antigen specificity of B cells of interest. Additionally, splice isoforms can be detected at the single cell level, which we have demonstrated by detecting *IGH* mRNA isoforms destined for antibody secretion or membrane-integration. RAGE-Seq also sequences receptors from all lymphocytes in a single reaction, including γδ T cells that are of increasing interest in infection and cancer immunology[42]. Finally, RAGE-Seq is compatible with DNA barcoded antibody technologies Abseq, CITE-seq, and REAP-seq[53–55], which are powerful tools for immunophenotyping, by allowing the additional measurement of cell surface proteins.

A limitation of RAGE-Seq lies in the low recovery of cell barcodes due to the higher error-rate of base-called nanopore sequencing data. Additionally, a large number of PCR cycles are required to generate sufficient material for nanopore sequencing, which can distort the distribution of cell barcodes and UMI sequences[56]. Here, we chose direct cell barcode matching for demultiplexing to reduce variability and false positives. We anticipate that better bioinformatics tools, such as those leveraging raw nanopore signal[57], will readily increase the recovery of antigen–receptor consensus sequences and the efficiency of cell-barcode recovery. A large number of TCR or BCR specific nanopore reads aligned to the majority of T or B cells suggests that the targeted capture approach employed by RAGE-seq can retain the information needed for antigen–receptor assembly. While we have relied on de novo assembly methods to generate splice isoform consensus sequences, we believe that detection and quantification of isoforms will be enhanced with the future identification of unique molecular identifiers (UMI) sequences from nanopore sequencing reads. Additionally, improved sequencing chemistry will reduce the error-rate of base-called nanopore data and limit the number of PCR cycles required for input.

While we have focused on antigen–receptor sequences, any transcripts of interest can be targeted using variations of RAGE-Seq, simply by changing the composition of the capture probe

library. A recent study performed long-read single cell sequencing to identify alternative isoforms in the mouse brain without targeted capture[58]. However, without the use of an enrichment strategy, the limited sequencing depth of long-read sequencing platforms results in only highly expressed genes being sampled and a high cost per cell. The high accuracy and single nucleotide precision achieved by RAGE-Seq will be particularly applicable for identifying somatic mutations in cancer, which could be applied to track the transcriptional consequences of subclonal mutations at single cell resolution. The adaptability of RAGE-Seq across multiple scRNA-Seq platforms and the flexibility to target a range of genes offers a new genomic toolkit for advanced single cell analysis.

## Methods

**Patient sample.** Treatment naive stage two triple negative breast cancer tissue and metastasized sentinel lymph node used in this work were collected under protocol X13-0133, HREC/13/RPAH/187. Human research ethics committee approval was obtained through the Sydney Local Health District Ethics Committee (Royal Prince Alfred Hospital zone), and site-specific approvals were obtained for all additional sites. Written consent was obtained from all patients prior to collection of tissue. Clinical data were stored in a de-identified manner, following pre-approved protocols. Tissue analysis was performed under protocol x14-021, LNR/14/RPAH/155.

**Single-cell suspension preparation and cell sorting.** Following surgical resection of tumor and lymph node (sentinel) from patient, samples were transferred in ice cold RPMI-1640 (Gibco) with 50% FCS to the laboratory to be processed. The tumor sample was cut into pieces approximately 1 mm³ in size and dissociated with the Tumor Dissociation Kit (Miltenyi Biotec). Lymph node was similarly processed however with digestion halted at 15 min. After washing twice with PBS containing 2% FBS, cells were passed through 70-μm strainers. The Jurkat T-cell line and Ramos B-cell line were cultured in RPMI-1640 with 10% FCS. Viable cells were enriched prior to scRNA-Seq by staining cells with DAPI (Invitrogen) followed by flow cytometric sorting using the Aira III Flow Cytometer (BD Biosciences). A gating threshold was used to omit red blood cells. Monocytes were flow sorted from human peripheral blood mononuclear cells (PBMCs) following staining with CD14-PerCP-Cy5.5 (BioLegend, Cat #325621) antibody (1/100 dilution). For Smart-Seq2 experiments, individual Jurkat cells were sorted directly into 96-well plates containing cell lysis buffer[59]. Prior to loading on the 10x Chromium instrument, cells were counted using a haemocytometer and the concentration of cells adjusted to ~1 × 10³ cells μL⁻¹. A viability of at least 90% for all samples were confirmed by trypan blue staining. Samples were handled on ice where possible.

**Droplet-based scRNA-Seq (10x Genomics).** scRNA-Seq libraries were prepared using the Chromium Single Cell 3′ v2 protocol (10x Genomics), aiming for recovery of 4000 cells for each sample. Briefly, single cells were encapsulated into droplets in the Chromium Controller instrument for cell lysis and barcoded reverse transcription (RT) of mRNA, followed by amplification, shearing and Illumina library construction. Two modifications were made to the protocol. Two extra PCR cycles were performed on top of the recommended number of cycles following the RT step. 30 μL of cDNA was used for library construction following full-length cDNA amplification, except for the tumor sample where 35 μL of cDNA was used. The remaining cDNA was used for targeted capture and Nanopore sequencing. An Illumina NextSeq 500 instrument (150 bps, paired-end) was used to sequence the scRNA-Seq libraries at a depth of >50,000 raw reads per cell.

**Antigen–receptor capture probe design.** A target enrichment library (Roche–NimbleGen) was designed by obtaining gene annotations of all functional V (*IGHV, IGKV, IGLV, TRAV, TRBV,* and *TRGV*), J (*IGHJ, IGKJ, IGLJ, TRAJ, TRBJ, TRGJ,* and *TRDJ*), and constant (*IGHA, IGHD, IGHE, IGHG, IGHM, IGKC, IGLC, TRAC, TRBC, TRGC,* and *TRDC*) TCR and BCR genes obtained from the IMGT database[60]. For each gene, genome coordinates of their corresponding exons were obtained from the GRCh38 primary assembly. Design of probes from target regions and synthesis was performed by Roche–NimbleGen using the SeqCap RNA Choice format with a maximum of five matches to the human genome. Sixty-six regions were removed from the final design due to being too small according to the NimbleDesign tool. In total 644 exons were targeted by the CaptureSeq array targeting ~128 Kb. A list of genes used for the capture array can be located in Supplementary Table 7.

**Targeted capture.** The remaining amplified full-length cDNA from droplet-based scRNA-Seq that was not used for Illumina library construction was used for targeted capture. Prior to capture, PCR was performed using KAPA HotStart HIFI ReadyMix (Kappa Biosystems) with 3 μM TSO primer (AAGCAGTGGTAT-CAACGCAGAGT) and 3 μM R1 primer (CTACACGACGCTCTTCCGATCT) and the following cycling conditions: 98 °C for 3 min; [98 °C for 20 s, 65 °C for 30 s,

72 °C for 1 min 30 s] × 5 cycles (cell lines) or × 20 cycles (primary cells); 72 °C for 3 min. Next, PCR products were purified using AMPure XP beads (Agencourt) and between 500 ng and 1 μg of amplified cDNA was used for targeted enrichment following the Roche–NimbleGen double-capture protocol as described in the SeqCap EZ Library support literature ("NimbleGen SeqCap EZ User's Guide [http://netdocs.roche.com/PPM/SeqCapEZLibrarySR_Guide_v3p0_Nov_2011.pdf] and "Double Capture Technical Note [http://netdocs.roche.com/PPM/Double_Capture_Technical_Note_August_2012.pdf]". Briefly, cDNA libraries were incubated overnight at 47 °C with probes and hybridization reagents (SeqCap EZ Accessory Kit v2 #07145594001; SeqCap EZ Hybridisation and Wash Kit #05634261001). Libraries were washed and hybridized a second time overnight for further enrichment. Universal hybridization enhancing (HE) oligo and index HE oligos were not included during hybridization. Following each round of hybridization and capture, PCR was performed using KAPA HotStart HIFI with 1 μM TSO primer and 1 μM R1 primer (instead of the TS-PCR oligos) with the following PCR cycling conditions: 98 °C for 3 min; [98 °C for 20 s, 65 °C for 15 s, 72 °C for 1 min 30 s] × 5 cycles (first round) or × 20 cycles (second round); 72 °C for 3 min. Post-capture cDNA library size ranged from 0.6 to 2 kb.

**Nanopore sequencing.** Targeted captured cDNA libraries were prepared for long-read sequencing using Oxford Nanopore Technologies' (ONT) 1D adapter ligation sequencing kit (SQK-LSK108), with the exception of one sample that used the 1D² adapter ligation kit (LSK-308). The latter was base called and considered as 1D for all subsequent steps. All samples were sequenced with R9.4.1 flowcells (FLO-MIN106), with the exception of 3/6 cell line samples that were loaded onto R9.5.1 (FLO-MIN107) flowcells (including the aforementioned LSK308 sample). Base calling was performed offline on a high-performance computing cluster using ONT's Albacore software pipeline (version 2.2.7). A list of samples, chemistries, flowcell identification numbers, and manufacturer software versions can be found in Supplementary Table 8.

**Smart-Seq2.** Smart-Seq2 scRNA-Seq was performed using the protocol described by Picelli et al.[59]. Briefly, single cells were sorted into cell lysis buffer containing 0. μL RNase inhibitor (Clontech), 1.9 μL Triton X-100 solution (0.2%), 1 μL dNTP mix (10 mM), and 1 μL oligo-dT primer (5 μM). Reverse transcription was performed containing 0.5 μL SuperScript II reverse transcriptase (200 U/μL, Invitrogen), 0.25 μL RNAse inhibitor (40 U/μL, Clontech), 2;μL Superscript II First-Strand Buffer (5×, Invitrogen), 0.25 μL DTT (100 mM, Invitrogen), 2 μL betain (5 M, Sigma), 0.9 μL MgCl₂ (100 mM, Sigma), 1 μL TSO (10 μM). Reverse transcription was carried out at 42 °C for 90 min, followed by 10 cycles of 50 °C for 2 min and 42 °C for 2 min. PCR was performed using KAPA HiFi HotStart ReadyMIX (KAPA Biosystems) with 28 cycles of PCR and the IS PCR primer reduced to 50 nM. Sequencing libraries were prepared using the Nextera XT Library Preparation Kit (Illumina) and sequencing was performed on the Illumina NextSeq platform (150 bps, paired-end) at ~1 million reads per sample. Following sequencing, reads were processed using the VDJPuzzle algorithm[13] to determine TCR sequences.

**Droplet-based scRNA-Seq data analysis.** Raw sequencing data were demultiplexed, aligned to the GRCh38 genome and UMI-collapsed using the Cell Ranger software (version 2.0, 10x Genomics). The raw gene-expression matrices were normalised and scaled using Seurat (v3.4)[61]. Quality control was performed on each dataset to remove poor quality cells. For the cell line dataset, cells that expressed less than 250 genes or less than 1000 UMIs were excluded, while for the lymph node and tumor datasets cells a threshold of less than 100 genes or less than 500 UMIs was used. Cells that contained more than 6% UMIs derived from mitochondrial genome were excluded from cell line datasets, and more than 10% mitochondrial UMIs for primary both tumor and lymph node. Doublets were identified as any cells expressing greater than 6500 genes or deviated more than ×4 of the median gene count within each cell type. Doublets were removed for the cell line dataset.

For each dataset a principle component analysis was performed on the variable genes using the Seurat workflow. Using the Jackstraw method[61], the first 40 principle components with a *P*-value < 0.01 was used for dimensional reduction. The resolution set for each tSNE analysis was determined using well known canonical marker genes and Seurat's FindAllMarkers function yielding an average expression for any particular cluster >2.0-fold higher than the average expression in other sub clusters from that cell type. Seurat's default Wilcoxon rank sum test was used for differential gene-expression analysis with a *P*-value < 0.01. Tumor and lymph node datasets were combined using Seurat's RunCCA to enable a comparative analysis. Cell cycle scoring was performed using scRNA-Seq cell cycle gene-expression scores from Tirosh et al.[62]. The V(D)J sequences of each cell were integrated into a Seurat object as metadata for gene expression and clonotype analysis. Raw non-collapsed or collapsed UMI count for each experiment was extracted from molecule_info.h5 file generated by Cell Ranger (version 2.0 10x Genomics) to determine the number of *TRAC* or *IGHM* reads or UMIs per cell.

**Demultiplexing nanopore sequencing data.** Base called fastq files were pooled for each biological sample and subjected to ad hoc demultiplexing using a direct sequence matching strategy (i.e., 0 mismatches and indels). Cell barcode sequences

(16 nt) were extracted from matched Illumina sequencing data produced by CellRanger software (version 2.0, 10x Genomics). Forward and reverse-complemented cell-barcode sequences were used to demultiplex the nanopore sequencing reads by scanning the first and last 200 nt of any read longer than 250 nt for an exact match. Following demultiplexing the 13 nucleotides downstream of the position matching a cell barcode were removed for each read. This was performed to ensure that (i) the 10 nt UMI sequence is removed from consensus assembly steps, and (ii) potential insertions are also removed (using the script rageMatch.py). The fastq headers were also modified to include barcode and UMI sequences post-demultiplexing. The number of total nanopore reads and demultiplexed reads for each sample is shown in Supplementary Table 1.

**De novo assembly and error correction.** As highlighted in Supplementary Fig. 1, demultiplexed reads were grouped into distinct fastq files and subjected to de novo assembly with Canu (version 1.7 r8737)[33] using assembly parameters: -Overlapper = minimap -batMemory = 28 -minReadLength = 200 -minOverlapLength = 100 -genomeSize = 15k. The distinct fastq file were then aligned to the CANU contigs with Minimap2 (version 2.10-r763-dirty)[63] using option -k 15 before being used to correct the consensus sequence using Racon (version 1.2.1)[34] with parameters: -w 200 -m 8 -x -1 -g -4. The Minimap2/RACON step was repeated a total of four times, after which the corrected consensus was subsequently 'polished' with ionic current data present in the raw nanopore sequencing output files (fast5) with Nanopolish (version 0.9.0)[35]. These steps were run on a Sun Grid Engine high-performance computing cluster and the associated scripts (quick_polisher.sh & nano_polisher.sh) can be found on the github repository.

**TCR and BCR assignment.** Nanopore polished fasta files containing consensus transcript contigs for each cell barcode were subjected to IgBLAST[64] alignment to determine V(D)J rearrangements and BLASTN alignment[65] to determine the Ig or TCR constant regions exons associated with the V(D)J. For each contig, separate IgBLAST for BCR and TCR were performed using IMGT germline gene reference datasets[60]. Amino acid sequences and location of CDR3 were defined by the conserved cysteine-104 and typtophan-118 based on the IMGT numbering system. IgBLAST parameters were default with the exception of returning only a single gene segment per V(D)J loci. Following the first round of IgBLAST, insertions and deletions (indels) in parts of the sequence that aligned to germline gene segments were corrected to their closest germline gene, and the IgBLAST step was repeated to generate indel corrected alignments. This indel correction was performed to overcome the impact of Nanopore sequencing errors on the reading frame of the V (D)J rearrangement which if left uncorrected can prevent the CDR3 from being determined accurately (See Supplementary Fig. 7). We predicted this indel correction would have a minimal effect on calling somatic hypermutations of the V region of BCRs since the majority of AID induced mutations consist of single nucleotide substitutions[36]. Text-based IgBLAST output was then parsed to tab-delimited summaries, calling gene segments, framework and complementarity determining regions and mismatches relative to germline gene segments.

The following filtering of BCR and TCR sequences were performed following parsing of tab-delimited summaries. Antigen–receptor sequences that were out-of-frame or that contained stop codons, termed non-productive sequences, were removed, unless stated otherwise. BCR sequences containing more than 40 mutations or TCR sequences with more than five mutations in their respective V gene segments were removed from the dataset. This filtering step was not performed for the analysis of point mutations in the V region of Jurkat in Supplementary Fig. 4b. If a cell was assigned two or more different TCR sequences with the same V and J genes but different CDR3 nucleotide sequences, the TCR sequence with the least number of mutations in the V region was assigned. If there were no differences in the number of V region mutations only the receptor sequence with the greatest number of reads used during de novo assembly with Canu was assigned. For cells assigned two or more different BCR sequences with the same V and J genes only the BCR sequence with the greatest assembly read coverage was assigned.

Expanded clones or shared clones across tissues were defined by more than one cell sharing the same V and J germline gene segments with identical CDR3 amino acid sequences for the T cells, and same V and J germline gene segments with 90% identical CDR3 nucleotide sequence for B cells. Expanded clones or shared clones across tissues were measured using either the same shared paired chains (e.g., BCR heavy and light chains, TCRα and TCRβ, TCRγ and TCRδ, or cells carrying the same TCRβ chain, where specified. MAIT-associated TCRs were identified based on the usage of *TRAV1-2* and one of the J segments *TRAJ33*, *TRAJ20*, and *TRAJ12*[44]. GEM-associated TCRs were identified based on the usage of *TRAV1-2* and *TRAJ9* gene segments[45].

**Assignment of splice isoforms.** To determine the spliced constant regions exons that were associated with the V(D)J rearrangement blastn was used to align each contig against the spliced reference exons. For the *IGHC*, both the membrane and secreted versions of each constant region were included. Tabular blastn output was parsed to call constant region for each contig using the criteria of greater than 95% coverage of the spliced constant region exons and percentage identity of more than

90%. A 90% identity threshold was used because contigs used for constant region calling were not corrected for insertions or deletions.

**Generation of clonal network.** The Ramos B-cell line undergoes constitutive somatic hypermutation[36] generating cells that carry different somatic point mutations within their BCR chains. To examine the relationships among the single B cells from the cell line, a germline unmutated ancestor (UA) was inferred by reversion of mismatches to the germline reference. Pairwise distances between the heavy and light chain sequences for each cell, and to the reference, were calculated by hamming distance. A network was built among the single cells and the UA where each node grouped all cells sharing the same heavy and light chain amino acid sequences and edges represent interactions between nodes with lowest pairwise hamming distance and edge weight is the hamming distance between the nodes. The network was drawn in Cytoscape[66] using the Prefuse Force Directed Layout.

**Nanopore read subsampling.** Read subsampling was performed on 200 Jurkat and 200 Ramos cells with each cell having no less than one thousand reads. The subsampling itself was performed with the sequence analysis toolkit, seqtk version 1.0-r72 (https://github.com/lh3/seqtk), using the sample command with a seed parameter of -s123. Subsampling was performed in a stepwise manner at increments of 1000, 500, 250, 100, and 50 read depths, with the resulting subsampled fastq the next input in later rounds of subsampling.

**Determining on-target nanopore alignments.** Alignment of nanopore reads to TCR and BCR genes was performed by the alignment program Minimap2 (version 2.3-r536)[63] to a custom reference fasta sequence (refs.fa, available in the associated GitHub repository) containing TCR and BCR constant region genes, then counted reads not flagged as unmapped, not primary or supplementary using SAMtools[67] version 1.7-2-gc6125d0 (with htslib 1.7-6-g6d2bfb7). Specifically, we employed the command: minimap2 -a -x map-ont refs.fa reads.fastq | samtools view -F 0 × 904 -c.

**Statistics.** Figure 2.c, a Pearson correlation was performed. Supplementary Fig. 5h, a paired Wilcoxon test was performed when comparing Jaccard similarity coefficient for the 250 most abundant genes chosen from within each respective cell type cluster.

**Reporting summary.** Further information on research design is available in the Nature Research Reporting Summary linked to this article.

## Data availability
Sequencing data generated in this study have been deposited in the European Nucleotide Archive under the primary accession code PRJEB28878. All other data are available from the authors upon reasonable request.

## Code availability
All scripts used for the computational pipeline of RAGE-Seq can be found at: https://github.com/KCCG/rageseq.

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

## Acknowledgements

We acknowledge the assistance of Chia-Ling Chan, Kate Harvey and Sunny Wu with single cell captures, David Koppstein with computational analysis, and Joseph Powell for stimulating discussions. This work was funded by The Kinghorn Foundation, The National Breast Cancer Foundation, John & Deborah McMurtrie, The Bill and Patricia Ritchie Foundation, and NHMRC Program Grant 1113904. We gratefully acknowledge the support of Dr Laurence Gluch, Associate Professor Charles Chan and Professor Sandra O'Toole in providing clinical samples for this study. G.A.A. is the recipient of a UNSW Postgraduate Award and A.S. is a Senior Research Fellow of the NHMRC.

## Author contributions

M.S., G.A.A., C.C.G., M.A.S., and A.S. conceived and designed the project. M.S., and G.A.A., designed the experiments and methodology. G.A.A. processed the clinical samples for droplet sc-RNA-Seq. M.S. performed the targeted capture experiments. K.B. and M.A.S. performed the nanopore sequencing. S.L.C., J.M.F., and M.A.S. analysed the nanopore sequencing data. K.J.L.J. performed the repertoire analysis. G.A.A. analysed the droplet scRNA-Seq gene-expression data. M.S. assisted with data analysis. F.L. analysed the Smart-Seq2 data. T.G.P., S.J., D.L.R., and J.B. advised on experiments and data analysis and assisted with interpretation of results. M.S., G.A.A., C.C.G., M.A.S. and A.S. wrote the manuscript, with input from all other authors.

## Additional information

**Competing interests:** M.S., G.A.A., C.C.G., S.L.C., J.M.F., K.J.L.J., M.A.S., and A.S. have filed a patent application covering some aspects of this work. The remaining authors declare no competing interests.

