## [Peer Review File · Nature Communications]

Reviewers' comments:

Reviewer #1 (Remarks to the Author):

Singh et al. in this manuscript have developed a pipeline to simultaneously uncover the antigen receptor transcripts sequences and genome-wide gene expression profiles at a single lymphocyte level, by combining sequence capture, nanopore sequencing and assembly of antigen receptor transcripts, with the commercial 10x 3' single cell RNA-seq platform. They have tried to validate the pipeline in immortalized T and B cell lines (Jurkat and Ramos), and primary breast tumor tissue and adjacent lymph node. This manuscript is very well written in language. Single cell profiling of T and B lymphocytes requires their receptor sequences as cell proxies or barcodes to track their origins and differentiation paths, and provide valuable information for cell affinity to antigens. Currently, full-length whole transcriptome sequencing based on single cell sorting to plates or high-density microfluidics (such as Takara ICELL8 system) prior to smart-seq, is able to profile the T and B cell receptor sequences and gene expression at the same time, and 10x also launches their V(D)J solution based on drop-seq. While the pipeline introduced in this manuscript has some technique novelties (such as hybridization based V(D)J enrichment and nanopore long-read sequencing), the authors should make a more comprehensive and subjective comparison over the existing methods, in terms of cost, sensitivities and applications. Specifically, some important questions and concerns regarding technological bias, sensitivities and sequencing errors should be addressed prior to its publication.

Major:

1. An important technological implementation in the pipeline is to use probe-based sequence capture instead of PCR-based method to enrich the V(D)J transcripts in single cells. However, the sensitivity in detecting low-frequency receptor transcripts is relatively low by sequence capture, demonstrated by the low recovery rate of paired TCR and BCR chains in both the cell lines and primary cells. Substantial PCR cycles are used in pre- and post-capture process to ensure enough materials for capture and sequencing. Moreover, hybridization kinetics may bring selection bias towards shorter fragment length (the length distribution of post- vs. pre-capture). The authors adopted their published protocol of RNACap-seq, however, recovering V(D)J transcripts at single-cell level is unique. The authors may want to comment and analyze more on the issues raised above.
2. Nanopore sequencing produce super-long reads, but the sequencing error rate is higher than short-read platform. This is critical in calculating the somatic hypermutations in single B cell. In the B cell clonal network analysis part, the authors mentioned that the mismatches to the germline sequences of Jurkat cells is 5.05% of TRAV and 2.8% of TRBV, this is an enormous number compared with other sequencing platform. What is the hypermutation rate of Ramos cells versus the expectation? Is there any difference between CDR and FR region? In Fig. 4, the mutation rate is very high in Ramos cells. In Fig. 5C some naïve B cells exhibit mutation rate higher than 4% or even 10%, and memory B cells also reveal higher mutation rate than expectation. The involvement of sequence errors would bring the mutation calculation and clonal network analysis unreliable.
3. In the introduction and discussion part, the authors present a brief comparison with SMART-seq and 10x V(D)J solution. I would suggest a more thorough comparison in a table with all V(D)J single cell platforms including ICELL8 system in terms of all aspects such as throughput, cost, sensitivity et al., and clarify the pros and cons of RAGE-seq over other methods and pipelines.

Minor,

4. Typo, fig 2a, "Jukat" should be "Jurkat"
5. Fig 2d, the length distribution of IgH transcripts contigs doesn't match well with the prediction. Please explain or discuss.
6. Supplementary Fig 4, it's very difficult to differentiate light and dark blue dots. Other colors preferred.

7. The authors should clarify the immunohistological status of the lymph node in the study (negative or positive/metastatic lymph node), which influences the activation environment.
8. Page 15, "...13 cell pairs with the same BCR sequence, the majority of which segregated in the naïve B-cell cluster...", this observation seems unexpected.
9. Page 18, "11/1771 TCR α chains, 7/1475 TCR β chains and 5/155 TCR **.." garbled.
10. In page 27, The latest version is Seurat v3.0 by now, please ensure the correct version of software.
11. In section "Data availability", Sequence is not found under accession PRJEB28878 in European Nucleotide Archive.
12. Supplementary fig.4, legend, line 3, "IGLV2-13" should be " IGLV2-14"
13. In either tumor or lymph node, do T or B cells in different subsets (eg., Teff and Treg) share the identical paired receptors?

Reviewer #2 (Remarks to the Author):

The paper from Singh et al focusses on using two technologies to analyse the same input DNA sample to investigate single cell transcriptomes including primary tumours. Exploring the best of both long and short range sequencing in a single method is extremely interesting and has potential in the future.

I do have some reservations about the work as presented which may be addressable by the authors. In some parts (particularly the first paragraph of the introduction) there are somewhat broad claims about cell diversity in humans being dependent on gene rearrangement and alternative RNA splicing. These statements lack precision and are not well referenced. What is meant about cell diversity here? Developmental diversity? Phenotypic diversity? That they are genetically diverse is tautologically true. More precise wording would help.

It's a shame that the authors do not provide a direct comparison with other methods which can avoid isolation of specific cell type populations in this manuscript. Most specifically I would have liked to see a comparison using the same material with the Single Cell V(D)J + 5' Gene Expression kit. Given the timing of this manuscript with the release of this commercial kit, this may not have been practical. The authors reference this method in the discussion and identify several potential improvements of RAGE-Seq over the method it is really only direct comparison that would prove these. The comparison to Smart-seq2 and VDJpuzzle is perhaps misleading as these older methods require single cell selection. Thus the costings presented in supplementary table 3 are skewed in favour of any method which can run on droplets rather than the limited number of cells presented here. These costings are not presented in an equivalent way either - the cost per experiment is only reported for RAGE-Seq and not Smart-Seq2? It appears that RAGE-Seq is being compared against methods which might no longer be considered optimal for these experiments.

My major concern with the manuscript is pipeline used for decoding the Nanopore reads themselves. The authors suggest that the majority of reads cannot be de-multiplexed (approximately 20% of the reads). However, only 42.9% of reads are on-target. Is there any skew in the proportion of on-target reads which are correctly de-multiplexed? Or is it the same proportion of reads in both on and off target?

I struggle a little here with exactly the methods employed by RAGE-seq for demultiplexing the reads. If I have followed the manuscript correctly, the cell barcodes are identified from Illumina data and then matched using simple identity matching. I was surprised at the simplicity of this approach. The authors comment in the discussion that more sophisticated methods might help and cite deepbinner as one method. However, given that this is trained on known barcode sequence its surely unlikely to work with UMIs? Presumably knowledge of the set of UMIs present in the Illumina data can be used to explore the observed sequences in the Nanopore sequence data. This

step is surely critical as correctly demultiplexed reads are fundamental for all subsequent analysis and polishing steps. I see no obvious validation that this simple approach is sufficient? Perhaps it is the subsequent analysis that proves this but ultimately 80% of the Nanopore data is lost in this experiment.

As the authors themselves note, the low level of successful demultiplexing is limiting as is the additional PCR cycles required.

As an aside - I looked at the `rageMatch.py` script. The naming of the main function isn't particularly helpful (line 83/85 in `rageMatch.py`)! I couldn't find the files "`quick_polisher.sh` and `nano_polisher.sh`" in the GitHub repository.

The authors suggest in the introduction that they "predicted that such approaches could also be applied to Nanopore reads generated from cDNA targeted capture...". This isn't surprising. However the choice of using de novo assembly on these reads is surprising. cDNA targeted capture should result in reads which align relatively easily due to at least one fixed end. Nevertheless, the method appears to work. However it is not a surprise that the de novo assembly approach retains full-length mRNA transcripts - surely these are what are being fed into the process? Indeed figure 2d shows peaks for each of the read groups at the appropriate length for the relevant full length mRNA transcripts.

Overall the analysis is reasonable but the comparator is mainly on cost and throughput and thus the data should be compared to methods which can also exploit droplet approaches. Alternatively, the authors should make clear that similar cost benefits would be obtained from such methods.

It is not really clear to me how RAGE-Seq might be of particular value for a comprehensive description of a human cell atlas. Such a project is likely to benefit from higher depth and not have such cost constraints?

Minor queries:

Typo in line beginning "A proportion of receptor cells were found..." first paragraph page 18.

The percentages reported in supplementary table 1 are not correct. As far as I can determine they are 18.7, 15.6 and 18.4?

Response to reviewers

Reviewer 1

Response to 1st major point:

“An important technological implementation in the pipeline is to use probe-based sequence capture instead of PCR-based method to enrich the V(D)J transcripts in single cells. However, the sensitivity in detecting low-frequency receptor transcripts is relatively low by sequence capture, demonstrated by the low recovery rate of paired TCR and BCR chains in both the cell lines and primary cells. Substantial PCR cycles are used in pre- and post-capture process to ensure enough materials for capture and sequencing. Moreover, hybridization kinetics may bring selection bias towards shorter fragment length (the length distribution of post- vs. pre-capture). The authors adopted their published protocol of RNACap-seq, however, recovering V(D)J transcripts at single-cell level is unique. The authors may want to comment and analyze more on the issues raised above.”

Thank you for raising the above point. We were not very clear at explaining the recovery of TCR and BCR chains in the manuscript. We do not believe that targeted capture significantly diminishes the yields of BCR and TCR recovery. With the targeted capture approach we achieve a large number of TCR and BCR specific reads assigned to a large proportion of T and B cells (see Supplementary Figure 2D and 3B). We believe the low recovery rate of paired chains is a result of the *de novo* assembler used to generate consensus sequences. See Supplementary Figure 3B, C, which shows cells assigned a large number of Nanopore reads but no receptor chain being assigned. We anticipate better bioinformatic Nanopore consensus tools will increase the recovery. We have revised the manuscript to address this in the Discussion section.

Response to 2nd major point:

“Nanopore sequencing produce super-long reads, but the sequencing error rate is higher than short-read platform. This is critical in calculating the somatic hypermutations in single B cell. In the B cell clonal network analysis part, the authors mentioned that the mismatches to the germline sequences of Jurkat cells is 5.05% of TRAV and 2.8% of TRBV, this is an enormous number compared with other sequencing platform.

-This statement in the manuscript was not written clearly, thank you for raising it. 5.05% and 2.8% refer to the number of Jurkat cells with >1 mutations in their V region and not the actual error rate. The mutation rate in the V region across all Jurkat cells is: TRAV, 0.095% and TRBV, 0.032%. In contrast the mutation rate of Ramos was 3.41% for IGHV and 2.13% for IGLV. This part of the manuscript has been revised in the results section under the “B-cell clonal network analysis” section.

“What is the hypermutation rate of Ramos cells versus the expectation?”

-The expected hypermutation rate of Ramos cells is difficult to compare since it is dependent on how long the Ramos cells are cultured for. The longer the cells are grown in culture, the more mutations they accumulate. Based on the reported per generation SHM rates for Ramos and assuming a 36 hours division rate, the expected SHM frequency after 28 days of culture would reach around 15.8 mutations sequence (in the IGHV region) or 5.32%. We have added to the manuscript the expected rate per generation based on reference #36.

“In Fig. 4, the mutation rate is very high in Ramos cells. In Fig. 5C some naïve B cells exhibit mutation rate higher than 4% or even 10%, and memory B cells also reveal higher mutation rate than expectation.”

A small number of naïve B cells in the lymph node exhibited mutation rates higher than 4% (13/401). Errors from Nanopore sequencing may explain this observation, however a greater variable is the accuracy of calling individual cells as naïve or memory by gene expression. As shown in the tSNE plot in Figure 5a, there are some cells classified as naïve that fall into the memory B cell cluster. Additionally, since naïve and memory B cells were determined by expression of *IGHD* (see Supplementary Figure 5), IgD+ CD27+ human memory B cells would be included in the naïve cluster. How naïve and memory B cells were called by gene expression was not clear in the manuscript. We have revised the text to include how these cell clusters were determined and noted that unswitched memory B cells may be included in the naïve B cell cluster. We found that the rate of somatic hypermutation of the memory B cells in the lymph node (IGHA1: 4.96%, IGH A2: 7.34%, IGHG1: 5.64%, IGHM: 3.56%) was comparable to reported deep IGH sequencing of memory B cells (PMID: 26311730) and single cell data (PMID:29659703). We have revised the manuscript to include this comparison to published reports.

Response to 3rd major point:

In the introduction and discussion part, the authors present a brief comparison with SMART-seq and 10x V(D)J solution. I would suggest a more thorough comparison in a table with all V(D)J single cell platforms including ICELL8 system in terms of all aspects such as throughput, cost, sensitivity et al., and clarify the pros and cons of RAGE-seq over other methods and pipelines.

We have included a table in the manuscript (Supplementary Table 7) that compares the costs, recovery and advantages/disadvantages to other high-throughput droplet platforms: 10X-VDJ, Scissor-Seq, and DART-Seq.

Response to minor points:

Thank you for these suggestions to improve the manuscript. All changes have been addressed or clarified as suggested.

Point 4: Typo has been addressed

Point 5: The smaller IgH transcript of ~900kb that was detected in Ramos cells is a splice isoform that skips the *IGHM* constant region CH1-CH3 exons. This small IgH isoform has been reported previously amongst a large proportion of IgH transcripts using long-read sequencing: PMID:25611855.

Point 6: The colours in this plot have been changed.

Point 7: The lymph node was a sentinel lymph node positive for tumour cells. This has been added to the methods section of the manuscript.

Point 8: The reason why there were fewer expanded clones in the memory B cell cluster than the naive B cell cluster could be due to our requirement of 90% CDR3 sequence similarity for calling a BCR clone. It is possible that expanded B cell clones in the memory cluster were not called due to a high mutation frequency. Additionally, there was almost double the number of naive B cells (444) than memory B cells (283).

Point 9: These typos have been addressed

Point 10: Version 2.3 of Seurat is the latest peer-reviewed release and was used for all analysis

Point 11: All raw sequences have been uploaded to ENA. Access with the following:
Username: Webin-50911.
Password: aTHqKJ99

Point 12: Typo has been addressed.

Point 13: Amongst the expanded T cell clones in the lymph node, paired TCR sequences from 10/13 clones were found on cells assigned the same cell type. The remaining 3 clones were assigned to: CD4 EM and CD4 CM 1, CD4 Treg and CD4 CM 1, CD4 EM and CD8 EFF.

Reviewer 2

"I do have some reservations about the work as presented which may be addressable by the authors. In some parts (particularly the first paragraph of the introduction) there are somewhat broad claims about cell diversity in humans being dependent on gene rearrangement and alternative RNA splicing. These statements lack precision and are not well referenced. What is meant about cell diversity here? Developmental diversity? Phenotypic diversity? That they are genetically diverse is tautologically true. More precise wording would help."

- Thank you for raising this, this sentence was vague and not clear. We refer here to cell phenotype diversity and how alternative RNA splicing and V(D)J recombination contribute to the evolution of cell type diversity. We have revised the manuscript so this statement is less of a broad claim and have included references.

"...I would have liked to see a comparison using the same material with the Single Cell V(D)J + 5' Gene Expression kit. Given the timing of this manuscript with the release of this commercial kit, this may not have been practical. The authors reference this method in the discussion and identify several potential improvements of RAGE-Seq over the method it is really only direct comparison that would prove these." We appreciate that a direct comparison with the single cell V(D)J + 5' gene expression kit would be valuable, however as the reviewer points out, experiments performed in this manuscript were performed prior to the release of the 10X kit.

"The comparison to Smart-seq2 and VDJ Puzzle is perhaps misleading as these older methods require single cell selection. Thus the costings presented in supplementary table 3 are skewed in favour of any method which can run on droplets rather than the limited number of cells presented here. These costings are not presented in an equivalent way either - the cost per experiment is only reported for RAGE-Seq and not Smart-Seq2? It appears that RAGE-Seq is being compared against methods which might no longer be considered optimal for these experiments."

Thank you for these suggestions. Currently, Smart-Seq2 is still widely used for generating antigen receptor sequences compared to recently developed droplet based approaches. Thus we feel that a direct comparison to RAGE-Seq was justified. While we appreciate that Smart-Seq2 is plate-based and RAGE-Seq droplet-based, a large number of experiments utilise both technologies for the same sample. For example :PMID:30283141. The cost per experiment cannot be calculated in the same way as RAGE-Seq because the number of cells sequenced using Smart-Seq2 is up to the user for each experiment.

While it is difficult to accurately estimate the cost of performing experiments on each platform, we have included a table that compares the estimated costs, recovery and advantages/disadvantages to other high-throughput droplet platforms: 10X-5' VDJ, Scissor-Seq, and DART-Seq.

“My major concern with the manuscript is pipeline used for decoding the Nanopore reads themselves. The authors suggest that the majority of reads cannot be demultiplexed (approximately 20% of the reads). However, only 42.9% of reads are on-target. Is there any skew in the proportion of on-target reads which are correctly demultiplexed? Or is it the same proportion of reads in both on and off target?”

Indeed, using the methodology we describe in the manuscript, about 20% of the reads are demultiplexed using direct barcode matching. We anticipate that new sequencing chemistries--such as the R10 pore, released to early access users--will increase these yields in the future. Despite this, RAGEseq remains cost effective compared to other techniques given the higher throughput and lower sequencing cost (see Supplementary Tables 3, 4 & 7).

With regards to on-target reads, 43-55% of all cDNA reads mapped to targeted regions, including reads that present low base calling qualities. This proportion increases by about 7% when only considering reads that were demultiplexed, which are less likely to contain reads with poor base calling scores. As a reminder to the reviewer, a 43% on-target rate corresponds to a ~13-fold enrichment over the background, as determined by 3' gene expression levels from the unenriched short read sequencing.

“I struggle a little here with exactly the methods employed by RAGE-seq for demultiplexing the reads. If I have followed the manuscript correctly, the cell barcodes are identified from Illumina data and then matched using simple identity matching. I was surprised at the simplicity of this approach. The authors comment in the discussion that more sophisticated methods might help and cite deepbiner as one method. However, given that this is trained on known barcode sequence its surely unlikely to work with UMIs? Presumably knowledge of the set of UMIs present in the Illumina data can be used to explore the observed sequences in the Nanopore sequence data. This step is surely critical as correctly demultiplexed reads are fundamental for all subsequent analysis and polishing steps. I see no obvious validation that this simple approach is sufficient? Perhaps it is the subsequent analysis that proves this but ultimately 80% of the Nanopore data is lost in this experiment.”

Since RAGE-Seq is a novel technology, we chose a simple and robust method of demultiplexing to reduce variability and false positives as much as possible. As mentioned by the reviewer, demultiplexing by direct sequence matching is sufficient to accurately demultiplex cell barcodes, substantiated by the method's ability to generate accurate consensus sequences of antigen receptor sequences. Indeed, about 80% of the nanopore reads are lost in the process, however, the strategy remains cost-effective in comparison to other long-read single-cell methods (see response to reviewer 1 and new supplementary table with cost comparison).

Inevitably, as ONT chemistry upgrades and improved software tools emerge, the direct barcode matching strategy employed herein will increase demultiplexing efficiency significantly. We refer to DeepBinner as an example of methods predicated on raw signal analysis, which have the potential to increase demultiplexing efficiency. Our intention is to demonstrate that targeted nanopore sequencing can be combined with short-read transcriptome profiling from thousands of single cells, which we believe is clearly and accurately presented in this manuscript.

It should be noted that the nanopore de-multiplexing rate of ~20% is underestimated because the final cell barcode list has gone through standard 10X quality control filtering. We found that only between 50-80% of Illumina reads make up the final cell barcode list that are used to de-multiplex that nanopore data. Thus there is a considerable amount of Nanopore reads that cannot be de-multiplexed. We have revised the manuscript to include a table of the percentage of Illumina reads that make up the final barcode list and comment that some nanopore reads will not be able to be demultiplexed. We have also added to the discussion the rationale for using a straightforward de-multiplexing strategy.

“As an aside - I looked at the `rageMatch.py` script. The naming of the main function isn't particularly helpful (line 83/85 in `rageMatch.py`)! I couldn't find the files `quick_polisher.sh` and `nano_polisher.sh` in the GitHub repository.”

Thank you for pointing this out. Those files have been added to the GitHub repository.

“The authors suggest in the introduction that they “predicted that such approaches could also be applied to Nanopore reads generated from cDNA targeted capture...”. This isn't surprising. However the choice of using de novo assembly on these reads is surprising. cDNA targeted capture should result in reads which align relatively easily due to at least one fixed end. Nevertheless, the method appears to work. However it is not a surprise that the de novo assembly approach retains full-length mRNA transcripts - surely these are what are being fed into the process? Indeed figure 2d shows peaks for each of the read groups at the appropriate length for the relevant full length mRNA transcripts.”

The reasons we deliberated on *de novo* assembly of the nanopore reads is as follows:

1. During targeted capture, PCR and nanopore sequencing, fragmentation of full-length cDNA transcripts can potentially occur. *De novo* assembly mitigates the presence of low frequency fragmented sequences.
2. The 90% accuracy of individual reads is problematic for alignments across antigen receptor loci, where hundreds of small interspersed exons with smaller sequences can cause alignment errors.
3. *De novo* assembly followed by consensus polishing improves the sequence accuracy from 90% to >99%. This is vital for producing highly accurate antigen receptor sequences which can be assigned to individual lymphocytes.

We have revised the text to address the reasons for using *de novo* assembly.

“Overall the analysis is reasonable but the comparator is mainly on cost and throughput and thus the data should be compared to methods which can also exploit droplet approaches. Alternatively, the authors should make clear that similar cost benefits would be obtained from such methods.”

As stated above we have included a table that compares RAGE-Seq to other high-throughput droplet-based platforms.

‘It is not really clear to me how RAGE-Seq might be of particular value for a comprehensive description of a human cell atlas. Such a project is likely to benefit from higher depth and not have such cost constraints?’

-Thankyou for this comment. It is a complicated discussion and not particularly relevant to this manuscript. We have changed this sentence in the discussion and have not included the reference to the human cell atlas.

Response to minor points:

Thank you for these suggestions to improve the manuscript. All changes have been addressed or clarified as suggested.

REVIEWERS' COMMENTS:

Reviewer #1 (Remarks to the Author):

Most of my concerns have been properly addressed. I am glad the authors have clarified the mutation rate issues. The manuscript is improved and deserves publication after minor editing. Supplementary table 7 is good. However, there are some inaccuracies, such as, 10x VDJ kit claims it can provide full length transcripts, at least for TCRs. VDJ kit can also provide BCR somatic hypermutations, at least at the region it sequences. Relative part in the discussion should be rephrased accordingly. Lastly, comparison with the Takara ICELL8 system is not provided, which can produce full length VDJ transcripts at a reasonable cost and with high throughput.

Xiao Liu, PhD
BGI-Shenzhen

Reviewer #2 (Remarks to the Author):

Overall, the authors have addressed the majority of my comments to my satisfaction.

I am still puzzled by the efforts at costing the experiments as shown in supplementary tables 4 and 7. In supplementary table 4, the sum of the cost per cell column for Rage-Seq is around \$4, yet in table 7 the estimated cost is only \$2.50 per cell. Table 7 quantifies the costs for the lymph node experiment - is this different to the costs as calculated in table 4? Really these should be consistent or the differences and reasons for the differences made clear. Cynically, one might conclude that the lowest cost per cell calculation is being shown in table 7?

The authors have not really addressed my query with respect to the cost of an experiment particularly clearly. I appreciate that the total cost of the experiment is dependent on the number of cells sequenced, but I think this should be clearly explained in the legend to the appropriate table.

In my original comments, I note that the percentages as shown in table S1 were incorrect. However, the corrected table now also has a change in absolute numbers with respect to the previous dataset. Has this change altered anything else in the manuscript?

The authors have clarified my questions re: on and off target reads. However, I find the additional clarification that Illumina barcodes are also lost in the manuscript confusing. I think the authors are suggesting that were they to be looking for the entire Illumina barcode set they would be able to demultiplex 25% of the Nanopore reads. However, this requires some thought and could be further clarified in the manuscript.

Response to referee comments

Reviewer #1

Most of my concerns have been properly addressed. I am glad the authors have clarified the mutation rate issues. The manuscript is improved and deserves publication after minor editing. Supplementary table 7 is good. However, there are some inaccuracies, such as, 10x VDJ kit claims it can provide full length transcripts, at least for TCRs. VDJ kit can also provide BCR somatic hypermutations, at least at the region it sequences. Relative part in the discussion should be rephrased accordingly. Lastly, comparison with the Takara ICELL8 system is not provided, which can produce full length VDJ transcripts at a reasonable cost and with high throughput.

In supplementary table 7 (Now 6) we refer to recovery of the full-length TCR/BCR transcript including the constant region. This has been clarified in supplementary table 6 as “Full-length mRNA”. As far as we can tell, the 10X kit does not reports full length transcripts, as parts of the constant region are lost during targeted enrichment of VDJ transcripts. See page 17 of the 5' VDJ technical sheet, link provided below.

https://assets.ctfassets.net/an68im79xiti/7uMbmhREPYMVgCzhcZvhsT/3bca7db9f9aabc1ec28046cb731ceef8/CG000207_ChromiumNextGEMSingleCellV_D_J_ReagentKits_v1.1_UG_RevA.pdf

Snippet of from the link above:

We have not seen any data or publications that the VDJ kit can report somatic hypermutation. While the ICELL8 systems claims to sequence TCRs from single cells, we could not find any publications or technical data that reports TCR recovery and therefore cannot include it in the table.

Reviewer #2

I am still puzzled by the efforts at costing the experiments as shown in supplementary tables 4 and 7. In supplementary table 4, the sum of the cost per cell column for Rage-Seq is around \$4, yet in table 7 the estimated cost is only \$2.50 per cell. Table 7 quantifies the costs for the lymph node experiment - is this different to the costs as calculated in table 4? Really these should be consistent or the differences and reasons for the differences made

clear. Cynically, one might conclude that the lowest cost per cell calculation is being shown in table 7?

The reason why the cost per cell is lower for the lymph node sample than the cell line sample (Sup Table 4) is because more cells were processed for the lymph node sample and a similar amount of reagents and sequencing were used for both samples. The cell line had 3,743 cells in the final data set while the lymph node had 6,027. We chose the lymph node sample for Table 7 (Now table 6) since we wanted to compare primary T and B cells. We have added to the Supplementary Table 7 (Now table 6) legend the reason explaining this difference.

The authors have not really addressed my query with respect to the cost of an experiment particularly clearly. I appreciate that the total cost of the experiment is dependent on the number of cells sequenced, but I think this should be clearly explained in the legend to the appropriate table.

Thank you for raising this point. We have explained in the legend of Supplementary Table 4 our explanation of why the cost of an experiment cannot be calculated for SmartSeq2 in the same way as 10X Chromium.

In my original comments, I note that the percentages as shown in table S1 were incorrect. However, the corrected table now also has a change in absolute numbers with respect to the previous dataset. Has this change altered anything else in the manuscript?

This has not changed any downstream analysis in the manuscript.

The authors have clarified my questions re: on and off target reads. However, I find the additional clarification that Illumina barcodes are also lost in the manuscript confusing. I think the authors are suggesting that were they to be looking for the entire Illumina barcode set they would be able to demultiplex 25% of the Nanopore reads. However, this requires some thought and could be further clarified in the manuscript.

This has been further clarified in the manuscript.